# In Vitro Mechanical Properties of a Novel Graphene-Reinforced PMMA-Based Dental Restorative Material

**DOI:** 10.3390/polym15030622

**Published:** 2023-01-25

**Authors:** Francesco De Angelis, Mirco Vadini, Matteo Buonvivere, Antonio Valerio, Michele Di Cosola, Adriano Piattelli, Virginia Biferi, Camillo D’Arcangelo

**Affiliations:** 1Unit of Restorative Dentistry and Endodontics, Department of Medical, Oral and Biotechnological Sciences, University of Chieti, 66100 Chieti, Italy; 2Department of Clinical and Experimental Medicine, University of Foggia, 71122 Foggia, Italy; 3School of Dentistry, Saint Camillus International University of Health and Medical Sciences, Via di Sant’Alessandro 8, 00131 Rome, Italy

**Keywords:** resin-based composites, PMMA, graphene, Vickers hardness, flexural strength, compressive strength

## Abstract

Recent studies suggest that the incorporation of graphene in resin-based dental materials might enhance their mechanical properties and even decrease their degree of contraction during polymerization. The present study aimed at comparing the three-point flexural strength (FS), the compressive strength (CS), and the Vickers hardness (VH) of a CAD/CAM poly-methylmethacrylate (PMMA)-based resin, a recently introduced graphene-reinforced CAD/CAM PMMA-based resin (G-PMMA), and a conventional dental bis-acryl composite resin (BACR). No significant differences (*p* > 0.05) were detected among the materials in terms of flexural strength. On the other hand, a mean flexural modulus value of 9920.1 MPa was recorded in BACR group, significantly higher compared to the flexural modulus detected for G-PMMA (2670.2 MPa) and for conventional PMMA (2505.3) (*p* < 0.05). In terms of compressive modulus (MPa) and compressive strength (MPa), BACR was significantly stiffer than PMMA and G-PMMA. Concerning VH measurements, a significantly increased hardness emerged comparing the BACR group (VH 98.19) to both PMMA and G-PMMA groups (VH 34.16 and 34.26, respectively). Based on the finding of the present study, the graphene-reinforced (PMMA)-based polymer herein tested was not superior to the conventional PMMA and seemed not able to be considered as an alternative material for permanent restorations, at least in terms of hardness and mechanical response to compressive stress. More research on the mechanical/biological properties of G-PMMAs (and on graphene as a filler) seems still necessary to better clarify their potential as dental restorative materials.

## 1. Introduction

Resin-based materials commonly used in dental practice can be divided into those based on dimethacrylates or bis-acryl/composite resins (such as bisphenol A-glycidyl dimethacrylate (Bis-GMA) and urethane dimethacrylate (UDMA)) and those based on monomethacrylates (or acrylic resins), which include poly-methylmethacrylate (PMMA) [1]. These materials have different chemical structures, mechanical properties, and polymerization kinetics [1,2,3].

Bis-acryl/composite resins (BACRs) are composed of an organic polymeric matrix, fillers (inorganic particles such as crystalline quartz, pyrogenic silica, glasses of barium, zinc, and strontium, and ceramic), coupling agents, pigments, catalysts, and inhibitors [4]. The basic chemical composition of dimethacrylates provides them with an appropriate mechanical behavior against applied stresses, as they have a rigid, cross-linked structure owing to the presence of highly viscous and voluminous multifunctional monomers that can cross-link with other polymeric chains. This cross-linking, combined with inorganic loading, makes these materials strong and easy to handle and to polish [5,6,7,8,9]. BACRs can withstand sufficiently high forces before breaking. All these properties make them suitable for both provisional and permanent restorations [10,11]. Once the stress is greater than the elastic deformation limit, BACRs tend to fracture rather immediately, instead of undergoing any important plastic deformation. Consequently, they can be described as brittle materials [7,8].

PMMAs are also amongst the most commonly used polymer in dental laboratories, dental clinics (for relining dentures and temporary crowns), and industry (fabrication of artificial teeth) due to their unique properties: low density, low modulus of elasticity, aesthetics, cost-effectiveness, ease of manipulation, and relatively fast manufacturing process [12]. PMMA is conventionally available in the form of a powder–liquid system. The powder contains a clear polymer (PMMA); however, additives such as pigments and nylon or acrylic synthetic fibers are added to adjust the physical properties and aesthetics. The liquid component contains a monomer of methyl methacrylate, along with cross-linking agents and inhibitors [12,13]. Over the last years, with advancements in computer-aided design/computer-aided manufacturing (CAD/CAM), CAD/CAM PMMA-based polymers have been introduced. They may offer mechanical advantages over conventionally polymerized PMMA resins thanks to a highly cross-linked structure and can be effectively used as alternative materials for long-term temporary prostheses [14]. Under load, unlike BACRs, PMMA-based materials undergo a progressive permanent plastic deformation phase before fracture, over which a gradual reduction in the load values is observed [13].

Graphene is one of the allotropic forms of carbon and consists of a single layer of sp2 hybridized carbon atoms arranged in a "honeycomb" structure. Graphene and its derivatives have many applications in the fields of science and technology due to their physical and chemical properties, such as electrical conductivity, transparency, biocompatibility, superior mechanical strength, and high surface area [15,16]. Some recent studies suggested that the incorporation of graphene in PMMA materials might further enhance resin’s mechanical properties and decrease the degree of contraction during polymerization [17,18,19,20]. Lee et al. reported that PMMA exhibited better antimicrobial-adhesion effects after incorporating graphene [21]. On the above bases, some manufacturers claimed the use of their graphene-reinforced PMMA-based material even for permanent prostheses [22]. To date, however, the available evidence supporting a mechanical superiority of graphene-reinforced dental restorative materials still seems weak [22,23,24]. In particular, there are only a few studies, to date, that aimed at comparing the in vitro mechanical properties of graphene-reinforced resin-based dental materials, justifying the need for further research in this perspective. Therefore, the present investigation aimed at comparing the three-point flexural strength (FS), the compressive strength (CS), and the Vickers hardness (VH) of a CAD/CAM PMMA-based resin, a recently introduced graphene-reinforced CAD/CAM PMMA-based resin, and a conventional bis-acryl-based nanohybrid dental composite. BACR was used as a control because the mechanical properties of bis-acryl composite resins are generally superior to PMMAs and they are unanimously considered suitable for permanent restorations [25,26]. The null hypothesis to be tested was that there were no statistically significant differences among the mechanical properties of the three tested materials.

## 2. Materials and Methods

Information about the materials under investigation and the abbreviations used in the present study to identify the experimental groups are given in Table 1.

### 2.1. Three-Point Bending Test

Samples to be subjected to the three-point bending test were made as follows.

Both in PMMA and in G-PMMA groups, prismatic-shaped samples, with dimensions equal to 2 mm × 2 mm × 25 mm, were milled out directly into the desired shape starting from a pre-polymerized wafer using the conventional CAD/CAM methodology (Figure 1). Finishing and smoothing were achieved using 180 μm grit sandpaper. In order to obtain a standardized level of finishing, all tools were worked by a single operator using round circular motions and uniform pressure.

In BACR group, resin samples were made by compressing the uncured conventional nano-hybrid composite material between two microscope slides with interposed polyacetate sheets, inside a prismatic steel mold, having an internal dimension of 2 mm × 2 mm × 25 mm [27]. To ensure the detachment of the sample, a thin layer of Vaseline was brushed on the internal surfaces of the mold. The material was then light-cured for 40 s on both sides with a high-power LED curing light (Celalux 3, © 2021 VOVO GmbH, Cuxhaven, Germany), following an overlapping technique. After curing, the prismatic samples were removed from the mold and finished as for PMMA and G-PMMA groups. They were then subjected to a 10 min heat curing cycle (LaborLux, Micerium) to ensure complete polymerization.

Ten specimens were manufactured in each group (*n* = 10).

All samples were subjected to loading until fracture in a universal testing machine (LR30K; Lloyd Instruments Ltd, Fareham, UK) equipped with a 500 N load cell (Figure 2).

Samples were placed in a test fixture for three-point bending test, built according to the NIST guidelines n. 4877, with a span distance of 20 mm. The crossbar of the machine was moved with a descent rate of 0.5 mm/min towards the upper surface of the specimen and the load/deflection curve was recorded with Nexygen-Ondio software (version 4.0, Lloyd Instruments) until specimen fracture. The load (N) and deflection (mm) were recorded. The flexural strength (σ_f_) (MPa) was calculated based on the following formula:σ_f_ = 3 F_max_ l/(2wh^2^) 

F_max_ = fracture load (N);

l = span distance (mm); 

w = width of the sample (mm);

h = height of the sample (mm).

Flexural modulus (E_f_) of elasticity (Mpa) was calculated as the slope of the load/deflection curve in the elastic deformation range of the sample, using the following formula:E_f_ = l^3^ F/(4wh^3^d) 

F = load (N);

d = deflection (mm). 

In each experimental group, means and standard deviations for flexural strength and flexural modulus were calculated.

### 2.2. Compressive Strength Test

Samples to be subjected to the compressive test were made as follows.

Both in PMMA and in G-PMMA groups, cylindrical samples with dimensions of 8 mm in height and 4 mm in diameter were made with the CAD/CAM method, milling them from a pre-polymerized wafer, and then finished using 180 μm grit sandpaper (Figure 3).

In BACR group, samples were made by stratifying, and then polymerizing (Celalux 3, VOCO, Cuxhaven, Germany) for 40 s, 4 composite layers of 2 mm in thickness, inside a cylindrical steel mold with a height of 8 mm and a diameter of 4 mm. To ensure the easy removal of the sample, a thin layer of Vaseline was brushed on the internal surfaces of the mold. Two microscope slides with interposed polyacetate sheets were used to ensure perfectly flat and smooth top and bottom surfaces. After polymerizing the last layer, the samples were removed from the mold and any flash material was removed with a disposable scalpel having a 15 blade. Subsequently, the samples were subjected to an additional 10 min heat curing cycle to ensure complete polymerization.

Ten specimens were manufactured in each group (*n* = 10).

Compressive strength was determined by subjecting each sample to a compressive load using a universal testing machine (LR30K; Lloyd Instruments Ltd, Fareham, UK) equipped with a 30KN load cell (Figure 4).

After placing the flat end of the specimens on the support plate, they were subjected to an axial load with a descent speed of 0.5 mm/min. The load/deflection curve was recorded with the Nexygen-Ondio software (version 4.0, Lloyd Instruments) up to the maximum load (N) capable of determining sample fracture or yield.

The compressive strength (σ_c_) (MPa) for each cylindrical sample was calculated according to the following formula:σ_c_ = F/A 

F_max_ = maximum load (N);

A = specimen cross-sectional area (mm^2^).

The compressive Young modulus (E_c_) was calculated as the slope of the load/deflection curve in the elastic deformation range of the sample, according to the following formula:E_c_ = Fh/(AΔh)

F = load (N);

h = initial specimen height of the (mm);

A = specimen cross-sectional area (mm^2^);

∆h = variation in the specimen height, within the limit of proportionality (mm).

In each experimental group, means and standard deviations were calculated for the compressive strength and for the compressive Young modulus.

### 2.3. Vickers Hardness Test

Samples to be subjected to VH measurement were made as follows.

Both in PMMA and in G-PMMA groups, cylindrical samples with dimensions of 2 mm in height and 4 mm in diameter were milled out of a pre-polymerized wafer, following a conventional CAD/CAM method, and then finished using 180 μm grit sandpaper (Figure 5).

In BACR group, composite resin was placed into cylindrical molds with a 4 mm diameter and 2 mm height. To achieve in all samples flat and smooth top surfaces, the uncured paste was placed inside the mold in slight excess and covered with a transparent polyester film, followed by a microscope glass. Pressure was then applied to displace the excess material, and light curing was performed through the glass for 40 s (800-mW/cm^2^ output), maintaining the curing unit tip in direct contact with the glass and perpendicular to the composite specimens. All specimens were subjected to a final 10 minute heat curing cycle. Ten specimens were manufactured in each group (*n* = 10).

VH readings were recorded on the top smooth surface of the specimens. Vickers indentations were produced by applying a 10 N load for 10 s using a universal testing machine with a 500 N load cell (Lloyd LR 30K, Lloyd Instruments) provided with a standard 1368 Vickers diamond indenter (item #17, Affri, Induno Olona, Varese, Italy) (Figure 6) [28].

Subsequently, scanning electron microphotographs (EVO 50 XVP LaB6, Carl Zeiss, Cambridge, UK) were taken at different magnifications in order to measure the linear extent of the indentation diagonals (mm) (Figure 7).

VH numbers were then calculated according to the following formula:VH = (1.854 × F)/[(d_1_ + d_2_)/2]^2^

where d_1_ and d_2_ are the measured diagonals (mm) and F is the predetermined applied load expressed in kilograms force (1.0204 Kg).

For each specimen, the mean value of three VH readings performed at approximately 2 mm distance from one another was used as raw datum. Mean VH values were calculated in each experimental group.

### 2.4. Data Analysis

In each one of the three experimental groups, means (and standard deviations) for flexural strength, flexural modulus, compressive strength, compressive modulus, and VH were calculated. After having determined that data were not normally distributed, results were statistically compared using the Kruskal–Wallis one-way analysis of variance on ranks and the Dunn’s method for all pairwise multiple comparisons. The level of α was set at 0.05 in all tests.

## 3. Results

The mean values and the standard deviations for the flexural strength (MPa), flexural modulus (MPa), compressive strength (MPa), compressive modulus (MPa), and VH achieved in the three experimental groups tested are summarized in Table 2.

As expected, a different elastic/plastic behavior was observed when approaching the maximum load (N), with both PMMA-based materials showing a significant extent of plastic deformation before fracture. From the statistical analysis, no significant differences (*p* > 0.05) were detected among graphene-reinforced polymer (119.4 MPa), conventional PMMA (113.5 MPa), and conventional bis-acryl-based nanohybrid dental composite (125.7 MPa) in terms of flexural strength. On the other hand, a mean flexural modulus value of 9920.1 MPa was recorded in BACR group, significantly different from the flexural modulus detected for the graphene-reinforced polymer (2670.2 MPa) and for conventional PMMA (2505.3) (*p* < 0.05). 

Statistically significant differences were detected among the materials also in terms of compressive modulus (MPa) and compressive strength (MPa). Even in the compressive test, within the range of elastic deformation and direct proportionality between load and deformation, conventional bis-acryl-based nanohybrid dental composite showed a higher stiffness, with a mean compressive modulus value (4089.8 MPa) significantly increased compared to that found for the graphene-reinforced polymer (1937.8 MPa) and conventional PMMA (2285.8) (*p* < 0.05). Moreover, graphene-reinforced polymer and conventional PMMA showed significantly reduced compressive strength mean values (94.2 MPa and 101.1 MPa, respectively) compared to BACR group (135.9 MPa) (*p* < 0.05).

Concerning VH measurements, the hardness values achieved in conventional PMMA and G-PMMA groups (VH 34.16 and 34.26, respectively) were not statistically different from one another, but a significant difference emerged comparing both PMMA groups to the BACR group (VH 98.19).

## 4. Discussion

Since its discovery in 2004 [29], many applications have been explored for graphene, ranging from electronic and optoelectronic devices to photoconductive materials. However, only in 2008 was graphene introduced for the first time in the field of biomedical sciences [30] and widely used in biomedical applications such as bioelectronics, bioimaging, drug delivery, tissue engineering, and, not least, dentistry [31].

When a reinforcing filler is added into a dental restorative material, the mechanical reinforcement is not only determined by the intrinsic mechanical properties of the filler, but is also strongly dependent on the interfacial adhesion quality between the filler and the surrounding matrix. Experimental efforts were recently made to produce a uniform dispersion of graphene nanofillers in polymer matrix through the change in graphene surface chemistry and its interaction with the polymer matrix [32,33,34]. This was usually achieved through hydrogen bonding or covalent functionalization. For the first time, Wand G. et al. demonstrated that the impact of surface functionalization and functionalization degrees of graphene on the interfacial adhesion of G-PMMA clarifies the importance of optimized chemical functionalization on the interfacial stress transfer and fracture mechanism at a microscopic level. Furthermore, the presence of oxidative debris in the as-made graphene oxide could of benefit to the formation of G-PMMA due to a good dispersion and strong interfacial interaction between graphene and polymer matrix [20]. According to previous studies, incorporation of graphene in PMMA materials might further enhance resin’s mechanical properties [17]. For all the above reasons, some manufacturers claimed the use of their graphene-reinforced PMMA-based material even for permanent prostheses [22].

Of course, graphene is not the only nanofiller that can be incorporated into the resin matrices of dental restorative materials. In fact, reinforcement by metal oxide nanoparticles, such as silicon dioxide (SiO2), titanium dioxide (TiO2), and zirconium dioxide (ZrO2), is crucial for improvement of the mechanical properties, such as, for example, wear resistance, flexural strength, and tensile strength, leading to enhanced durability of the restoration [35,36,37,38,39,40]. However, Helal et al. recommended incorporating (SiO2) particles cautiously in PMMA denture teeth, as they may decrease wear resistance [41]. Despite the effects of different nanofillers on the mechanical properties of resin-based materials having already been studied [42,43,44], it could be interesting to update such research, including also graphene-reinforced resins. In this study, in order to assess the possible superiority of graphene-reinforced dental restorative materials, the flexural strength, the compressive strength, and the Vickers hardness of a CAD/CAM PMMA, a graphene-reinforced CAD/CAM polymer, and a conventional bis-acryl-based nanohybrid dental composite were evaluated. Those mechanical properties are among the most commonly used to compare the in vitro performance of dental restorative materials [45,46]. The clinical relevance of these tests lies in the information about the fracture resistance and the rigidity of a material [47], since recent published systematic reviews showed that fracture is one of the most frequent reasons for dental restorative material failure [48,49,50,51,52]. This is particularly true for flexural strength, Vickers hardness, and flexural modulus [53,54].

The conventional nano-hybrid composite resin showed, as expected, mechanical properties typical of an elastic but brittle material which, after a first phase of elastic deformation, undergoes sudden failure [55]. According to the analysis of flexural strength observed by Lee at al., PMMA-based materials, in turn, proved to be extremely less rigid materials compared to BACR, as they were capable of undergoing significant plastic deformation before breaking [56]. Alamgir et al. [57] observed a greater resistance to deformation of the PMMA–graphene composite, with a higher value of elastic modulus than PMMA itself. The present study confirmed a slightly increased flexural modulus for G-PMMA compared to conventional PMMA but without any statistically significant difference. Taking into account the statistically similar flexural strength recorded for PMMAs and BACRs, the increased flexural elasticity of PMMA-based materials (i.e., lower flexural modulus) could be seen as an advantageous aspect in terms of material response to the flexural loads generated during chewing, especially when dealing with the dilemma of implant abutment rigidity under mechanical stress, as already highlighted by Magne et al. [58].

Based on the compressive strength results, it clearly emerged that the maximum loads recorded on both conventional PMMA and graphene-reinforced polymer led to a specimen plastic deformation (thus, they should be better considered as “yield loads” instead of fracture loads). However, the yield loads of both PMMA and G-PMMA were significantly reduced compared to the maximum loads leading to nano-hybrid composite specimen fracture. A too low yield resistance could represent a problem for a permanent restorative material, as it could undergo undesired plastic deformation over time, even when subjected to physiological and sub-critical compressive loads.

Even concerning the VH test results, the integration of graphene did not increase the hardness compared to conventional PMMA, and both PMMA-based materials showed significantly decreased VH values compared to BACR group.

As for the clinical relevance, in contrast with some recent research [14], the present findings seem to suggest that the specific graphene-reinforced (PMMA)-based polymer herein tested is not superior to conventional CAD/CAM PMMA polymers, at least in terms of hardness and mechanical response to compressive stresses. Thus, the present data do not support extending the clinical indications of graphene-reinforced PMMAs beyond the clinical indications of conventional CAD/CAM PMMAs and suggest to keep on considering them as effective long-term interim materials instead of alternative materials for permanent restorations.

Among the limitation of the present research, it should be underlined that only commercially available dental materials were used and manufacturers did not disclose the amount of graphene added to the graphene-reinforced polymer herein tested, because the composition is protected by a patent. It was not specified the exact procedure used to include/bind the graphene into the PMMA polymer. Previous studies reported a graphene content for resin-based dental materials ranging from 0.024% wt/wt to around 2.52% wt/wt [34,59] and enhanced mechanical properties were recorded even for the minimal graphene concentration of 0.024% [33,60]. Due to the lack of mechanical reinforcement observed for the G-PMMA tested in this study, we assumed that the graphene content was still too minimal, probably in order to avoid any detrimental impact on the final aesthetic. In fact, graphene is intrinsically dark and tends to confer a greyish color to the materials [61]. In addition, some studies have shown a dose-dependent effect on the biocompatibility and toxicity of graphene and its derivatives. Wang et al. showed that the toxicity of graphene oxide (GO) to fibroblast cells was low when the concentration of GO was lower than 20 μg/mL, whereas the cytotoxicity of GO increased when the concentration was up to 50 μg/mL. Wang et al. investigated the cytotoxicity of GO in mice and the results demonstrated that, when the concentrations of GO were 0.1 and 0.2 mg, no toxicity was detected, while, with the increase in concentration to 0.4 mg, chronic toxicity was observed [62]. More precise information about manufacturing procedures (including exact graphene content) should be hopefully disclosed in order to better understand a possible correlation between the actual graphene concentration and the ultimate material mechanical properties and biocompatibility. In that respect, one limitation of the present study was that no attempt was made to investigate the composition and grain size of graphene added into PMMA by means of a scanning electron microscopy (SEM)/X-ray diffractometry (XRD) analysis [60] or through Raman spectroscopy [23,63], and this could be an interesting subject for further research.

Another weakness of this in vitro research was that it failed to replicate the complexity of intraoral conditions (including the influence of materials on cells) and tried to predict the in vivo reliability of a graphene-reinforced restorative material just from its in vitro mechanical properties. Some research seemed to show antimicrobial potentials of graphene-reinforced nanomaterials [21,64,65], such as the antiadhesive effect against microbial species in artificial saliva observed by Lee et al. [21]. The effects of graphene on bacteria structure, metabolism, and viability depend on the materials’ concentration, time of exposure, physical–chemical properties, as well as on the characteristics of bacteria used in the tests [64,65]. The presence of sustained antiadhesion properties in graphene-reinforced PMMA suggests its potential usefulness as a promising antimicrobial dental material for dentures, orthodontic devices, and provisional restorative materials [21], but these aspects were not investigated in the present study.

As a future prospective direction, further investigations could be planned on graphene-reinforced PMMA prototypes in order to estimate the maximum acceptable amounts of graphene that do not compromise the aesthetic properties. Afterwards, further studies would clarify whether any increase in graphene concentration (within the limits of aesthetic acceptability) could positively improve the mechanical and antimicrobial properties of graphene-reinforced PMMA-based materials.

## Figures and Tables

**Figure 1 polymers-15-00622-f001:**
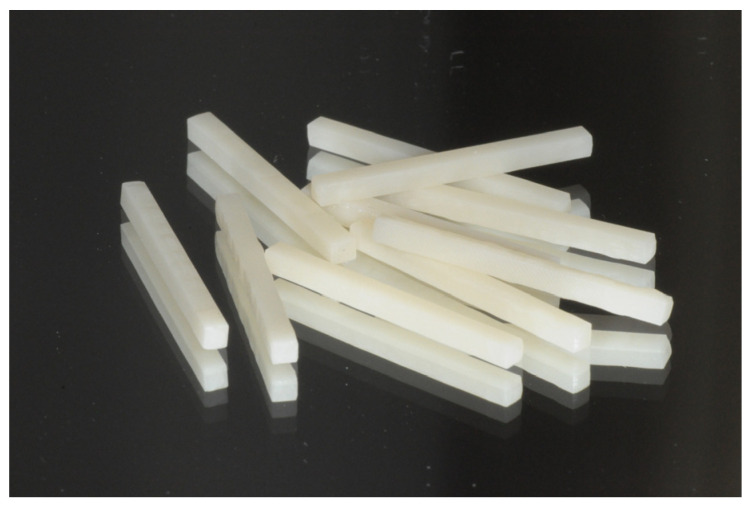
G-PMMA prismatic-shaped specimens after the finishing procedures.

**Figure 2 polymers-15-00622-f002:**
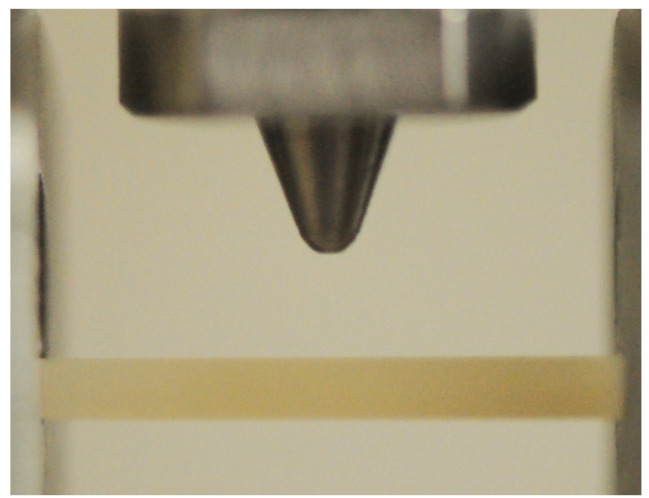
A prismatic-shaped specimen from the G-PMMA group, ready to be subjected to the three-point bending test.

**Figure 3 polymers-15-00622-f003:**
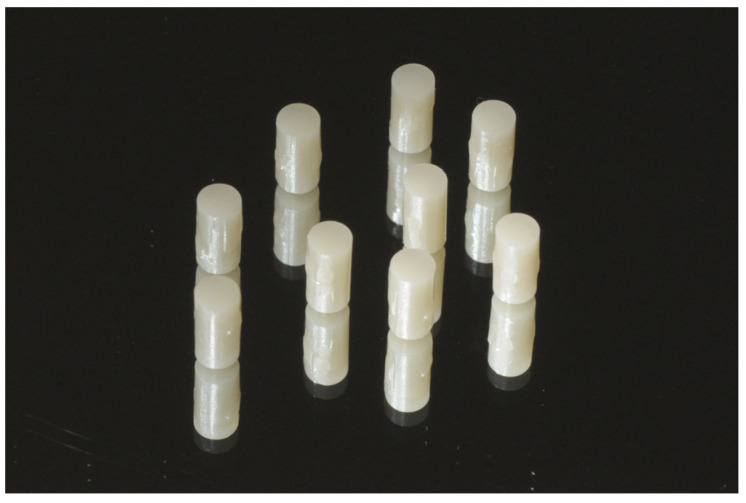
G-PMMA cylindrical samples.

**Figure 4 polymers-15-00622-f004:**
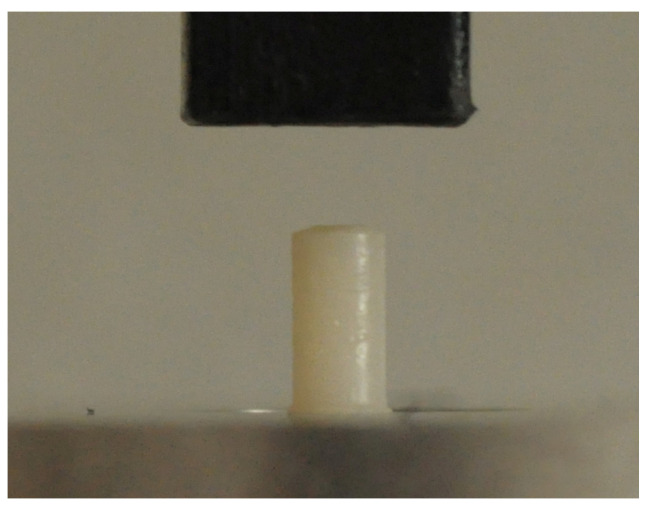
G-PMMA specimen ready to be subjected to compressive load in a universal testing machine.

**Figure 5 polymers-15-00622-f005:**
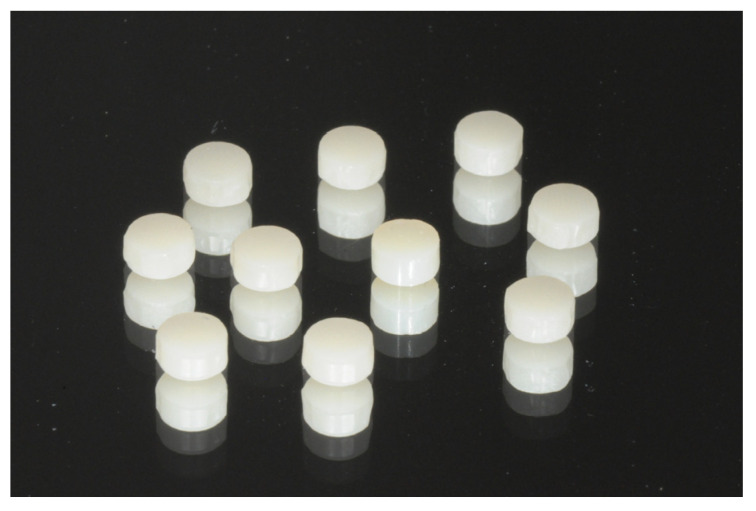
PMMA and G-PMMA cylindrical samples.

**Figure 6 polymers-15-00622-f006:**
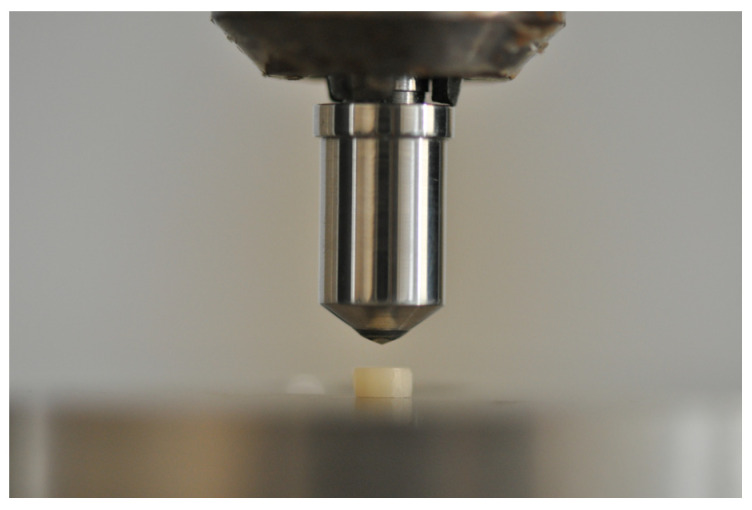
Sample subjected to Vickers indentation.

**Figure 7 polymers-15-00622-f007:**
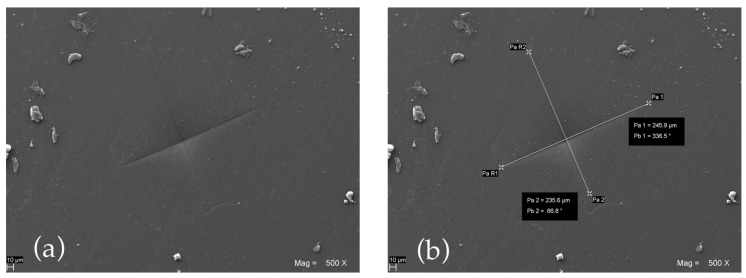
Scanning electron micrograph showing a VH indentation (**a**) and the measurement of its diagonals (**b**) performed on a specimen of G-PMMA.

**Table 1 polymers-15-00622-t001:** Summary of the materials tested.

Experimental Group	Type of Material	Material Trade Name	Batch Number	Manufacturer
Conventional PMMA	CAD/CAM Polymethylmethacrylate	Multilayer PMMA	85196	Dentsply Sirona, Roma, Italy
G-PMMA	CAD/CAM Graphene-reinforced Polymethylmethacrylate	G-Cam	L18101120161	Andromeda Nanotech, Lesignano de’ Bagni, Italy
BACR	Bis-acrylate composite resins	Enamel Plus HRi Biofunction	2022000987	Micerium, Avegno, Genova, Italy

**Table 2 polymers-15-00622-t002:** Mean values and standard deviations for flexural strength (MPa), flexural modulus (MPa), compressive strength (MPa), compressive modulus (MPa), and VH. Different superscript letters indicate statistically significant differences.

Experimental Group	Flexural Strength (MPa)	Flexural Modulus (MPa)	Compressive Strength (MPa)	Compressive Modulus (MPa)	VH
Conventional PMMA	113.5 ^a^ (13.3)	2505.3 ^b^ (486.1)	101.1 ^b^ (3.6)	2285.8 ^b^ (50.3)	34.16 ^b^ (3.67)
G-PMMA	119.4 ^a^ (9.0)	2670.2 ^b^ (199.7)	94.2 ^b^ (2.7)	1937.8 ^c^ (48.6)	34.26 ^b^ (2.12)
BACR	125.7 ^a^ (19.5)	9920.1 ^a^ (783.2)	135.9 ^a^ (20.0)	4089.8 ^a^ (438.7)	98.19 ^a^ (8.89)

## Data Availability

The data presented in this study are available on request from the corresponding author.

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
