# Peer review of "In Vitro Mechanical Properties of a Novel Graphene-Reinforced PMMA-Based Dental Restorative Material"

_polymers, 2023, doi:10.3390/polym15030622_

Round 1

Reviewer 2 Report

The manuscript entitled “In vitro mechanical properties of a novel graphene-reinforced PMMA based dental restorative material” by Angelis et al has compared the mechanical properties of three resin-based composites including CAD/CAM Polymethylmethacrylate, CAD/CAM Graphene-reinforced Polymethylmethacrylate and Bis-acrylate composite resins based on 3-point Flexural Strength (FS), the Compressive Strength (CS) and the Vickers hardness (VH). Overall, this work has its own merit in attempting to evaluated the mechanical properties mainly including compressive strength and hardness of a commercially available CAD/CAM Graphene-reinforced Polymethylmethacrylate compared to two conventional resin-based composites. However, the rationale of “(PMMA)-based polymers, even when reinforced with graphene, seem not yet able to be alternative materials for permanent restorations, at least in terms of hardness and mechanical response to compressive stress.” Should be proposed with caution, and the composition of this commercially available product, although closed, was not validated with some phase analysis including XRD and SEM. These major concerns along with other shortcomings in other sections required the manuscript for more discussion or detailed experiments about the reasonability for clinical practice before publication. The following points should be considered to improve the significance of this work.

Major concerns and comments:

Major concerns and comments:

1.The major result of this study was that there was no significant difference in mechanical properties between three resin-based composites including CAD/CAM Polymethylmethacrylate and CAD/CAM Graphene-reinforced Polymethylmethacrylate in terms of flexural modulus, flexural strength, compressive strength and hardness except compressive modulus. Based on this result, the authors concluded that “(PMMA)-based polymers, even when reinforced with graphene, seem not yet able to be alternative materials for permanent restorations, at least in terms of hardness and mechanical response to compressive stress.” However, there is only one study concerning the mechanical properties of PMMA reinforced with graphene cited which found an enhancement in mechanical properties of PMMA after incorporation of graphene. Numerous studies entitled “Tuning the interfacial mechanical behaviors of monolayer graphene/PMMA nanocomposites”, “Morphological and mechanical properties of graphene-reinforced PMMA nanocomposites using a multiscale analysis” and “Preparation of silanized graphene/poly (methyl methacrylate) nanocomposites in situ copolymerization and its mechanical properties” were not included for discussion. Moreover, the CAD/CAM Graphene-reinforced Polymethylmethacrylate used in this study was commercially available while its composition information is not available. This conclusion of the present study on only one product should be drawn with caution.

2. Although, the authors informed that “Manufacturers did not disclose the amount of graphene added to the graphene-reinforced polymer herein tested, because the composition is protected by a patent.”. However, some analysis including XRD and SEM is required to verify the composition and grain size of graphene added into PMMA.

3. In the “introduction” section, the background information was not enough for the novelty of this study. And the reason why chose BACRs as another control group was not well elucidated, especially for a clinical perspective.

4. In the “discussion” section, the results of this study were not thoroughly discussed, the underlying mechanism that whether the mechanical enhancement is provided by graphene was not well discussed.

5. In the “discussion” section, besides graphene, there are still many additions incorporated into resin-based composite for mechanical improvement and the comparison between these additions and graphene is required further discussion.

Minor comments:

1. In the “discussion” section, Line 302-304, no discussion about the clinical significance of these mechanical properties for resin-based restoration.

2. In the “discussion” section, Line 332-334, besides esthetics, the biocompatibility should also be considered.

Decision: Major revision.

Round 2

Reviewer 1 Report

Although innovation is rather low, the authors made the appropriate corrections and improve specific aspects. This version could be published.

Reviewer 2 Report

The authors has well answered the mentioned questions